# Study on the Performance of Deep Red to Near-Infrared pc-LEDs by the Simulation Method Considering the Distribution of Phosphor Particles

**DOI:** 10.3390/mi15081035

**Published:** 2024-08-15

**Authors:** Chenghang Li, Zikeng Fang, Ying Yan, Henan Li, Xiang Luo, Xuyue Wang, Ping Zhou

**Affiliations:** State Key Laboratory of High-Performance Precision Manufacturing, Department of Mechanical Engineering, Dalian University of Technology, Dalian 116024, China; chenghang@mail.dlut.edu.cn (C.L.); fangzikeng@mail.dlut.edu.cn (Z.F.); 201961166@mail.dlut.edu.cn (H.L.); luoxiang0920@163.com (X.L.); wbzzd@dlut.edu.cn (X.W.); pzhou@dlut.edu.cn (P.Z.)

**Keywords:** DR-NIR pc-LED, spectral power distribution, fluorescent film, phosphor distribution

## Abstract

Effectively utilizing deep red to near-infrared (DR-NIR) phosphors to achieve the optimal performance of NIR phosphor-converted white LEDs (DR-NIR pc-wLEDs) is currently a research hotspot. In this study, an optical model of DR-NIR pc-wLEDs with virtual multilayer fluorescent films was established based on the Monte Carlo ray-tracing method. Different gradient distributions of the particles were assigned within the fluorescent film to explore their impact on the optical performance of pc-LEDs. The results show that, for the case with single-type particles, distributing more DR-NIR particles far from the blue LED chip increased the overall radiant power. The distribution of more DR-NIR particles near the chip increased the conversion ratio from blue to DR-NIR light. The ratio of the 707 nm fluorescence emission intensity to the 450 nm excitation light intensity increased from 1:0.51 to 1:0.28. For multiple-type particles, changes in the gradient distribution resulted in dual-nature changes, leading to a deterioration in the color rendering index and an increase in the correlated color temperature, while also improving the DR-NIR band ratio. The reabsorption caused by the partial overlap between the excitation band of the DR-NIR particles and the emission band of the other particles enhanced the radiant power at 707 nm. Distributing DR-NIR phosphor particles closer to the chip effectively amplified this effect. The proposed model and its results provide a solution for the forward design of particle distributions in fluorescent films to improve the luminous performance of DR-NIR pc-wLEDs.

## 1. Introduction

According to the definition by the American Society for Testing and Materials (ASTM), near-infrared (NIR) light is typically defined as invisible light in the range from 780 to 2526 nm. However, in practical applications [1,2,3,4], due to similar characteristics, the NIR-I wavelength range is often extended to around 700 nm, covering a portion of the visible light spectrum (deep red, DR). Phosphor-converted LEDs (pc-LEDs) are one of the primary methods for obtaining NIR light and have been widely used in non-destructive testing [5,6], night vision lighting [7,8], plant growth [9,10], bioimaging [11,12], and health lighting [13,14]. NIR pc-LEDs, which consist of NIR phosphors and blue LED chips, have advantages such as small size, long lifespan, adjustable spectrum, and low cost, making them a focal point of research. As a key component of NIR pc-LED devices, NIR phosphors directly determine photoluminescence efficiency and spectral performance. Research on broadband NIR phosphors that can be efficiently excited by visible light (particularly blue light) is just beginning and their types are limited. Therefore, the exploration of broadband-emitting NIR phosphors that can be effectively excited by blue light is urgently required for the development of NIR pc-LED applications. Cr^3+^ with a 3d^3^ electronic configuration is considered to be an ideal NIR emission center. However, most Cr^3+^-doped broadband NIR phosphors exhibit low luminous efficiencies and poor thermal stabilities. Numerous researchers have focused on developing NIR phosphors with both high quantum efficiency and stability [15,16,17,18,19].

The material properties of near-infrared phosphors are fundamental to ensure the luminous performance of NIR pc-LEDs. However, for pc-LEDs, downstream manufacturing processes, such as coating, drying, and encapsulation of the fluorescent film, also significantly affect the luminous performance. The distribution of phosphor particles within the fluorescent film is a critical factor. Extensive research has confirmed that the sedimentation distribution [20,21,22,23], thermal convection distribution [24,25,26], centrifugal distribution [27], and local distribution [28] of phosphor particles within a fluorescent film significantly affect the spectral power distribution, correlated color temperature (CCT), and angular color uniformity (ACU) of pc-LEDs. In addition, when multiple types of phosphor particles are dispersed, the relative distribution positions of the different particles can result in varying luminous effects. Currently, extensive pre-experiments are often conducted to determine the optimal conditions before production, to ensure that the manufactured pc-LEDs satisfy the required luminous performance. Alternatively, post-production grading and selection are performed based on performance indicators. However, these methods are based on empirical accumulation and cannot provide effective guidance for designing the distribution of particles within fluorescent films. In addition, the research and preparation costs of phosphors, particularly efficient NIR phosphors, are relatively high. These methods inevitably lead to the wastage of rare-earth phosphor resources, increase the manufacturing costs of LEDs, and go against the economic principles of LED production.

Therefore, the use of optical modeling methods to evaluate the characteristics of the fluorescent conversion layer and its impact on the optical performance of pc-LEDs has become a current research focus. The Monte Carlo ray-tracing (MCRT) method has been widely reported and proven to be an effective modeling approach for analyzing the light path and evaluating the optical performance of various optical systems [29,30,31,32]. Sommer et al. [33]. simulated phosphor sedimentation in pc-LEDs using the commercial software package ASAP (Advanced Systems Analysis Program), based on the MCRT method. They found that the color temperature and flux output were highly sensitive to the phosphor distribution. Hu et al. [22] studied the effects of phosphor sedimentation in a silicone matrix on the light extraction efficiency (LEE), correlated color temperature (CCT), angular color uniformity (ACU), and light intensity distribution curve (LIDC) based on experimental observations and three-dimensional Monte Carlo ray-tracing simulations. The aforementioned results provide a solid methodological foundation for studying the performance of pc-LEDs through optical modeling. However, owing to the diverse applications of NIR pc-LEDs, there are instances where a single NIR band is used for detection and imaging, as well as scenarios where it is combined with phosphors of other wavelengths to create a more continuous spectrum for white-light illumination. In the existing modeling studies, the dispersion of multiple phosphors in pc-LEDs has rarely been reported, and research involving the use of NIR phosphor particles has recently begun.

In this study, to expand the application of the MCRT method and fill the current research gap regarding particle distribution in near-infrared fluorescent conversion layers, especially for multiple types of particles, an optical model of NIR pc-LEDs based on the MCRT method was established, considering the refraction, reflection, scattering, absorption, and conversion processes occurring in virtual multilayer fluorescent films. By assigning different gradient concentrations to the virtual multilayer fluorescent film model, the impact on the optical performance was explored for two cases: a single type of phosphor and multiple types of phosphors. This study focuses on exploring particle distribution forms that are more conducive to optimal emission spectrum realization through modeling, given a predetermined optimal phosphor particle volume fraction.

## 2. Methods

The fundamental principles underlying the modeling of pc-LED optical structures based on the Monte Carlo ray-tracing (MCRT) method are illustrated in Figure 1. First, a random number distribution similar to the light distribution of the blue LED chip was generated, with each random number corresponding to a ray emitted by the chip. The interaction of each ray with the geometric structure of the LED packaging, including scattering, reflection, and refraction, as well as the physical interactions between the ray and fluorescent particles, such as ray absorption and fluorescence conversion, were then considered until the ray reached the detection plane, thereby completing the entire travel process. The statistical solution of the ray distribution could be obtained by analyzing the results of a large number of such processes.

### 2.1. Modeling of the Blue LED Chip

The blue LED chip was 1 mm in length, 1 mm in width, and 0.05 mm in thickness. Because light was only emitted from the top surface, it was considered to be a surface light source. To ensure the accuracy of the Monte Carlo method, 10^8^ non-sequential rays were randomly generated, and the spatial distribution of the ray intensity followed the Lambertian cosine law, as expressed in Equation (1).
(1)$$I_{\theta }=I_{n},$$
where *I*_n_ represents the normal intensity of the rays emitted from the light source surface and *I_θ_* denotes the intensity of the rays at an angle *θ* to the normal direction. The spectral distribution of the blue LED chip followed a Gaussian distribution, as expressed in Equation (2), with a central wavelength of 450 nm (*μ* = 450 nm) and a full width at half maximum of 18 nm (*σ* = 7.64 nm).
(2)$$f_{\mathrm{in}}\left(\lambda \right)=\frac{1}{\sigma \sqrt{2\pi }}\exp\left(-\frac{{\left(\lambda -\mu \right)}^{2}}{2\sigma^{2}}\right).$$

The reflectance of the upper surface of the blue chip was set to 0.3, which controls the probability of rays reflecting after incident on the surface.

### 2.2. Modeling of the Fluorescent Film

The fluorescent film consisted of phosphor particles dispersed in optical silicone, and was modeled as a cube with a width and length of *w*_F_ = 1.2 mm and a height of *h*_F_ = 0.6 mm. In this study, it was modeled as a virtual 10-layer structure to facilitate the assignment of different particle volume fractions to each layer, as shown in Figure 1. It is assumed that there are no interfacial effects between the layers, indicating a transmittance of 1 and reflectance of 0. The reflectance coefficient of all side walls of the film was set to 0. This does not align with the actual structure where LEDs have reflecting cups and other components, but this simplified setting has a negligible impact on the comparative analysis of the spectral results of each simulation groups in this study. The total volume *V*_0_ of the fluorescent film is evenly divided into 10 layers, and the volume of each layer *V_i_* and the volume fraction of fluorescent particles *φ_i_* in each layer follow Equation (3).
(3)$$V_{0}\varphi_{0}=\sum_{i=1}^{10}V_{i}\varphi_{i},$$
The term *φ_i_* represents the percentage of the total volume of particles in the *i*-th layer relative to the total film volume of that layer. The optical silicone was assumed to be a homogeneous material with a constant refractive index of 1.41. The behavior of light in the fluorescent film layer adheres to the law of energy conservation; that is, the radiant flux Փ follows Equation (4):(4)$$\Phi_{\mathrm{in}}=\Phi_{\mathrm{out}}.$$

When light reaches the phosphor particles, it could be absorbed or not. Thus, Equation (4) can be written as follows:(5)$$\Phi_{\mathrm{in}}=\Phi_{\mathrm{scat}}+\Phi_{\mathrm{refl}}+\Phi_{\mathrm{conv}}+\Phi_{\mathrm{unco}}.$$

The occurrence of absorption was determined using the normalized absorption spectrum power distribution of the fluorescent particles. Whether the absorbed rays undergo fluorescence conversion is jointly determined by the normalized excitation spectrum power distribution and the emission spectrum power distribution. Rays that undergo fluorescence conversion are isotropically emitted. The rays that were not absorbed would be reflected or scattered by the particles. The reflection process and associated power loss follow Fresnel’s law. Because phosphor particles are approximately spherical and their diameter is larger than the incident wavelength (*d* > 0.1 *λ*), scattering follows the Mie scattering principle. This process considers the particle size distribution of the phosphor, as well as the refractive index and extinction coefficient of the optical silicone material and phosphor particles. The method for calculating the scattering angle distribution of rays through particles based on the Mie scattering theory has been widely reported [34,35,36] and will not be detailed here. Based on the Mie scattering theory, the mean free path *l* for the volumetric scattering of rays in the fluorescent film can be calculated using Equation (6). The mean free path is the average distance that a ray travels through the silicone material without striking the phosphor particles.
(6)$$l=\frac{1}{n\left(\sigma_{\mathrm{scat}}+\sigma_{\mathrm{abs}}\right)}$$
where *σ*_scat_ is the scattering cross-sectional area; *σ*_abs_ is the absorption cross-sectional area; and *n* is the density of the particle number, which can be calculated by the particle volume fraction *φ*, particle diameter *d*, and the defined volume of the film *V* in the simulation:(7)$$n=\frac{6V\varphi }{\pi d^{3}}$$

The mean free path can be used to calculate the probability distribution and determine the distance for each simulated ray propagation event.

### 2.3. Modeling of the Detection Plane

The detection plane, measuring 5 mm long and 5 mm wide, was placed parallel to and 10 mm above the top surface of the fluorescent film. The main workflow is illustrated in Figure 2. First, whether the ray intersects the detection plane was determined. If there is an intersection, then the wavelength and power information of the ray are collected. The collected rays were classified based on their wavelength intervals. Finally, the total power within each wavelength interval is calculated.

### 2.4. Simplified Settings of the Model

This study was conducted based on the following assumptions and simplifications.

The blue chip is defined to emit light only from its upper surface, with no light emission from the other four sides.The reflecting cup structure was not defined, and none of the simulation experiments in each group considered the reflecting cup, ensuring that each group had the same reflection conditions on both sides of the phosphor film (even if the reflectance was 0). This ensures that the relative changes in the spectrum distribution due to differences in particle distribution between groups are comparable.In the propagation path of the ray, all other propagation media not explicitly mentioned above were considered homogeneous, with a refractive index set to 1.

## 3. Results and Discussion

### 3.1. Validation Experiment with a Single Type of Phosphors

The phosphor particles used in this part of the simulation experiment were BaY_2_Al_2_Ga_2_SiO_12_:Cr^3+^ (Cr-707) provided by the Changchun Institute of Applied Chemistry. Weihong et al. [37] reported on the luminescence properties of DR-NIR phosphor particles. The photoluminescence excitation (PLE) and photoluminescence (PL) spectra and particle size distributions of the phosphor particles are shown in Figure 3. The PLE and PL spectra were measured using the fluorescence spectrometer (F-7000, Hitachi, Japan). As shown in Figure 3a, the excitation spectrum of Cr-707 particles has two peaks, one in the blue light band at 430 nm–450 nm and another in the 580 nm–630 nm range. The particle size distribution was measured using the laser particle size analyzer (BT-9300HT, Bettersize, Dandong, Liaoning, China). The external quantum efficiency of the Cr-707 phosphor is 30.3%. The results of the simulation experiment were compared using the spectra from the above literature. In the simulation, a uniform particle distribution was used, meaning that each of the ten virtual layers had an equal particle volume fraction *φ* of 10%.

As shown in the comparison of the normalized spectral distributions in Figure 4, the spectral distribution obtained through simulation using the pc-LED optical model established in this study agreed with that obtained from the literature [37]. The errors in the blue LED chip emission peak (450 nm) and phosphor emission peak (707 nm) were within the acceptable limits. These results indicate that the established optical model is effective and reliable for predicting the emission spectra of the pc-LEDs.

In addition to the uniform distribution of the particle volume fractions across the ten virtual layers, ten other gradient distributions were designed. Five of these gradient distributions (Gradient A to Gradient E) are shown in Figure 5, approximating the sedimentation and accumulation of phosphor particles towards the blue LED chip during the preparation and packaging of the fluorescent film. The other five gradient distributions (Gradient A–I to Gradient E–I) are the inverses of the five gradient distributions shown in Figure 5. All ten gradient distributions, as well as the uniform distribution, satisfy Equation (3) to ensure the comparability of the simulation results across different groups.

The simulation results for the normalized spectra with different particle volume fraction distributions in the virtual layered fluorescent film are shown in Figure 6. Compared with the uniform particle distribution (black curve in Figure 6a), the gradient distribution of the phosphor particles caused a significant overall shift in the spectral radiant power intensity. Furthermore, the direction of the shift caused by the accumulation of phosphor particles toward the blue LED chip and far from the blue LED chip was the opposite. When phosphor particles accumulate towards the blue LED chip, the overall spectral radiant power decreases, and the higher the accumulation concentration near the blue LED chip, the greater the reduction, as shown by the orange curves in Figure 6. When phosphor particles accumulate far from the blue LED chip, the overall spectral radiant power increases. The higher the accumulation concentration far from the blue LED chip, the greater the increase. This indicates that the probability of ray reflection or backward scattering near the chip decreases when the phosphor particles are distributed far from the chip. Light propagation occurred more frequently near the top of the film, causing more rays to be emitted from the top surface of the fluorescent film and reach the detection plane, thereby enhancing the overall radiant power. This phenomenon supports the finding that remote-phosphor coatings can enhance the luminous efficiency of LEDs [38,39]. However, the results in Figure 6b show that although the overall radiant intensity increases, the ratio of the 707 nm emission peak to the 450 nm emission peak decreases as the concentration of phosphor particles accumulating far from the chip increases. The ratio of the emission peak intensity at 707 nm to that at 450 nm decreased from 1:0.28 to 1:0.51, a reduction of approximately 45.1%. This is due to the refraction of the 450 nm rays from the blue LED chip within the film near the chip, which caused the 450 nm rays to escape through the edges and directly reach the detection plane. The aforementioned process reduced the proportion of 450 nm rays that were absorbed by phosphor particles and converted into 707 nm rays. These results indicate that even for LEDs that utilize only DR-NIR phosphor particles for fluorescence conversion, changing the concentration distribution in the fluorescent film significantly affects the final performance of the LED. In practical applications, fluorescent conversion films typically contain multiple types of phosphor particles that enable the LED spectra to cover a broader range of wavelengths. Therefore, cases involving multiple types of phosphor particles require further investigations.

### 3.2. Performance of DR-NIR pc-wLED with Multiple Types of Phosphors

For full-spectrum white LEDs that include deep red to near-infrared wavelengths (hereinafter referred to as DR-NIR pc-wLEDs), it is crucial to comprehensively evaluate their emission spectral performance to meet application requirements in health lighting, medical, and other fields. In addition to phosphor particles emitting at 707 nm (BaY_2_Al_2_Ga_2_SiO_12_:Cr^3+^, Cr-707), it is necessary to disperse additional phosphor particles with various emission wavelengths within the fluorescent film. In this study, three types of particles with emission wavelengths of 490 nm (BaSi_2_O_2_N_2_:Eu^2+^, Eu-490), 552 nm (Y_3_Al_5_O_12_:Ce^3+^, Ce-552), and 630 nm (CaAlSiN_3_:Eu^2+^, Eu-630) were selected. The overall volume fraction ratio of the four types of particles, *φ*_Cr-707_:*φ*_Eu-490_:*φ*_Ce-552_:*φ*_Eu-630_, was set to 5:1:2:7.5 The preliminary simulation experiments confirmed that the spectrum exhibited good continuity at this ratio. The PLE and PL spectra and particle size distributions of the phosphor particles are shown in Figure 7. The phosphor particle parameters used in this part of the simulation experiment were provided by the Changchun Institute of Applied Chemistry. The PLE and PL spectra were measured using the fluorescence spectrometer (F-7000, Hitachi, Japan). In the model, the PLE spectrum of the particles was used to determine the probability distribution of particles being excited at a given wavelength after absorbing light in that wavelength. The PL spectrum was used to represent the probability distribution of emitting light at various wavelengths after excitation. The particle size distribution was measured using the laser particle size analyzer (BT-9300HT, Bettersize, Dandong, Liaoning, China). When the phosphor particles were uniformly distributed in each layer of the fluorescent film, the volume fraction of Cr-707 was 5% per layer. The volume fractions of Cr-707 for the other gradient distributions were proportionally calculated, as shown in Figure 5. The other three types of phosphor particles were uniformly distributed. The benefit of altering only the concentration of Cr-707 while maintaining the uniform concentration of the other particles is that it allows for a rapid assessment of how changes in the relative position of Cr-707 to other particles affect the Cr-707 emission spectrum. If the distribution of the other three particles were also changed simultaneously, it would increase the coupling between the results, thereby complicating the analysis.

The simulation results for the addition of Eu-490, Ce-552, Eu-630, and Cr-707 particles to the fluorescent film are shown in Figure 8. The simulation results indicate that the addition of Eu-490, Ce-552, and Eu-630 compensates for the missing wavelength bands of the DR-NIR pc-wLED to some extent, thereby enhancing spectral continuity. Similar to the results in Figure 6, when Cr-707 accumulated at a position far from the chip, the overall spectral radiant power increased. Furthermore, the degree of enhancement increased with the increasing accumulation concentration, as shown in Figure 8a. The results shown in Figure 8b demonstrate that as Cr-707 accumulated farther from the blue LED chip, the relative intensities of the emission peaks changed. The peak value of the 450 nm blue emission was nonlinear. As the distribution of Cr-707 particles shifted from the position far from the chip towards the position near the chip, the peak intensity of the 450 nm blue light first increased, reaching a maximum point when the distribution was uniform, and then began to decrease. The relative change trends of the 490 nm and 552 nm bands are similar to those of the 450 nm band. For the 630 nm and 707 nm bands, the changes were monotonic. The ratio of emission peak intensity at 707 nm to that at 630 nm increased from 0.79:1.0 to 0.99:0.88, an increase of approximately 42.4%.

To more clearly characterize the relative changes in each peak of the spectral power distribution, spectral deconvolution was performed using the “Peak Deconvolution” app in Origin software. The reliability of the fitting process was checked by monitoring the *R*^2^ coefficient, determining the most suitable function form for modeling the spectrum. Trial-and-error analysis revealed that the Asymmetric double Sigmoidal function [40,41,42] (Equation (8)) best fits the emission spectrum of the phosphor particles in this study, and the spectrum of the blue LED was referenced using Equation (2).
(8)$$y=y_{0}+A\frac{1}{1+e^{-\frac{x-x_{c}+w_{1}/2}{w_{2}}}}\left(1-\frac{1}{1+e^{-\frac{x-x_{c}-w_{1}/2}{w_{3}}}}\right)$$
where *x* is the wavelength in the unit of nm; *y* is the intensity at *x*; *A* is the amplitude parameter; *x*_c_ is the center wavelength; *w*_1_ is the width; *w*_2_ and *w*_3_ are the skew parameters (or shape parameters). The baseline *y*_0_ was set to zero. The fitting parameters and fitting results for the emission spectra of the four types of phosphor particles are listed in Table 1. From the results of *RSS* and *R*^2^, the fitting performance is relatively good. The spectral fitting curves for the four types of particles are shown in Figure 8c. By comparing the fitted curves with the experimentally obtained emission spectra, it is evident that the fitting results are satisfactory.

Furthermore, using the aforementioned fitting parameters, the spectral power distributions for each group in Figure 8b were deconvoluted. The results are presented in Figure 8d. The results of Figure 8d are basically consistent with those of Figure 8b. It can be seen more clearly that, as Cr-707 particles accumulate closer to the chip, the relative radiation power at wavelengths other than 490 nm decreases compared to 707 nm. The relative intensity decreases are most pronounced in the Ce-552 and Eu-630 wavelength bands. This indicates that the accumulation of Cr-707 particles near the blue LED chip increased the relative intensity of the 707 nm emission peak, which is consistent with Figure 6b and Figure 8b. Additionally, this enhancement of the emission peak is also related to ray reabsorption occurring within the fluorescent film. A portion of the rays emitted at the 630 nm band by the excited Eu-630 particles were reabsorbed by the Cr-707 particles. A small portion of Ce-552 particles and the blue light chip also contributed to this effect. The significant decrease in the peak intensity at 630 nm observed in Figure 8b confirmed this point. This is due to the overlap between part of the excitation band of the Cr-707 particles and part of the emission band of the Eu-630 particles. In addition, Figure 8d shows that the change in the distribution of Cr-707 particles also affects the emission of Ce-552 and Eu-490 phosphor particles. The variation in Ce-552 emission is similar to that of Eu-630, except that the amplitude of the variation is not as significant as that of Eu-630. This can be explained by the fact that the excitation band of the Cr-707 phosphor covers only a small part of the emission band of the Ce-552 phosphor, and, in this part, the excitation efficiency of Cr-707 is relatively low. As for the Eu-490 phosphor, the changes in its relative emission intensity were relatively small, as seen in Figure 8d. As the Cr-707 particles accumulate towards the direction of the chip, the relative intensity of Eu-490 generally shows a trend of initially increasing and then decreasing. In other words, the relative emission intensity of Eu-490 is highest when the Cr-707 phosphor is uniformly dispersed in the film. Since the emission band of the Eu-490 phosphor is part of the excitation band of the other three particles, its variation is the result of the coupling of the other three particles, as well as the blue LED chip.

To evaluate the color performance of the DR-NIR pc-wLED, parameters such as the Correlated Color Temperature (CCT) [43], Color Rendering Index (CRI) [44,45], and Color Quality Scale (CQS) [46] were calculated. The results for CCT, CRI, and CQS are shown in Figure 9. As the particles accumulated from a position far from the chip towards nearer to the chip, the CRI value monotonically decreased from 95.62 to 92.11, while the CCT value monotonically increased from 3434 K to 3940 K. The variation trends of these two parameters indicate that the distribution of phosphor particles in the fluorescent film affects the color quality of the DR-NIR pc-wLEDs. Specifically, as shown by the CRI, the accumulation of particles towards the chip results in a decrease in the color quality of DR-NIR pc-wLEDs, and, the higher the accumulation concentration, the lower the quality. However, considering the CQS values, it initially shows an increasing trend, reaching a maximum value at Gradient D, after which it started to decline. The CRI and the CQS showed almost an inverse trend, and this is because the CRI cannot evaluate the color rendering performance of this type of spectrum for colors close to saturated red. CQS was proposed as an improvement and replacement for the CRI and is gradually being adopted for evaluating the color quality of LEDs. Therefore, if only CQS is used to evaluate the quality of DR-NIR pc-wLEDs, it implies that there exists an optimal particle distribution form, namely Gradient D.

Therefore, to comprehensively evaluate performance, new metrics must be introduced. For light sources used for specific wavelength applications, such as DR-NIR pc-wLEDs, it is crucial to assess whether the relative content of the DR-NIR band satisfies the usage requirements. Hence, the defined Band Ratio Index of *λ*_1_ to *λ*_2_ (*BRI_λ_*_1–_*_λ_*_2_) can be calculated using Equation (9), and the schematic diagram is shown in Figure 10a. The wavelength range in the numerator represents the target band to be examined and evaluated, while the range in the denominator can be set according to the emission range of the pc-LED. In this study, the emission spectrum of Cr-707 was divided into three wavelength ranges for analysis: 600–707 nm, 707–830 nm, and 600–830 nm, respectively. The results are shown in Figure 10b.
(9)$$BRI_{\lambda_{1}-\lambda_{2}}=\frac{\sum_{\lambda_{1}}^{\lambda_{2}}f_{\mathrm{exc}}\left(\lambda \right)\Delta \lambda }{\sum_{\lambda =380}^{830}f_{\mathrm{exc}}\left(\lambda \right)\Delta \lambda }$$

The variation of *BRI*_707–830_ shows an overall increasing trend, indicating that the closer the Cr-707 phosphor particles accumulate towards the chip, the higher the proportion of the DR-NIR band in the final output. However, the *BRI*_600–707_ wavelength range shows a gradual decreasing trend, which is due to this range covering part of the emission spectrum of Eu-630, including its central emission wavelength. During the transition from gradient E–I to gradient E in particle distribution, the relative radiative intensity of Eu-630 is in a declining state. Although the enhancement of the radiation intensity of Cr-707 phosphor particles did not reverse the downward trend of 600–707 nm, the decrease degree was reduced from the result. Taken all together, *BRI*_600–830_ is still in an increasing trend. Thus, for the selection of the *BRI*_λ1–λ2_ parameter band, it is necessary not only to consider the emission band of the selected phosphors but also to confirm whether it can independently reflect the radiation intensity change of the phosphors. Compared with *BRI*_600–707_, *BRI*_707–830_ is more suitable for evaluating whether the relative content of DR-NIR in pc-wLED spectra in this study can meet the application requirements.

The results indicate that the gradual accumulation of Cr-707 particles towards the blue LED chip leads to a decrease in certain performance aspects of the DR-NIR pc-LEDs, such as CRI, while improving other aspects, such as CQS and *BRI*_707–830_. Therefore, evaluating the color quality of DR-NIR pc-wLEDs requires finding a balance point that primarily depends on application requirements. To achieve the target metrics based on the application requirements, the distribution of phosphor particles in the fluorescent film must be carefully designed and manufactured. In particular, in cases where multiple phosphor particles need to be dispersed, it is crucial to fully consider the relative positioning of phosphor particles with respect to the excitation light source as well as the relative distribution between different phosphor particles. Although the 10-layer phosphor film model set in this study is difficult to achieve with current packaging technology, its guidance on gradient distribution is practical. The 10 layers can be simplified to a smaller number of layers that are feasible with current technology based on the gradient suggestions. In this way, the absorption, scattering, and reflection processes of light among different particles can be better utilized to achieve favorable light color performance. During the coating, drying, and encapsulation processes of the film, it is essential to appropriately control the phosphor particle distribution and maintain consistency between the films to obtain uniform light output in the DR-NIR pc-wLED products.

## 4. Conclusions

This study established an optical model of DR-NIR pc-wLEDs based on the Monte Carlo ray-tracing method. The effects on the spectral power distribution and color quality of DR-NIR pc-wLEDs were studied using a virtual multilayer fluorescent film model to control the distribution of phosphor particles. The following conclusions were drawn through simulations and comparative analyses.

When only near-infrared phosphor particles (Cr-707) were dispersed in the fluorescent film, the accumulation of Cr-707 particles far from the blue chip enhanced the overall radiant power of the DR-NIR pc-wLED, whereas the accumulation near the chip increased the relative intensity of the near-infrared band.For cases in which multiple phosphor particles were dispersed, the partial overlap between the excitation wavelength of Cr-707 and the emission wavelength of the other particles (Eu-630) led to reabsorption. Distributing Cr-707 phosphor particles closer to the blue LED chip than Eu-630 exacerbates the reabsorption, significantly enhancing the ratio of emission peak intensity at 707 nm to that at 630 nm increased from 0.79:1.0 to 0.99:0.88, an increase of approximately 42.4%.The performance changes caused by the redistribution of phosphor particles were dual in nature. The transition of phosphor particles from accumulation at a position far from the blue LED chip to accumulation near the chip results in a decrease in the CRI and overall radiant power, but simultaneously leads to an increase in CQS, and there exists a distribution gradient (Gradient D) that makes CQS optimal.The Band Ratio Index (*BRI_λ_*_1–*λ*2_) was proposed to characterize the relative content ratio of the near-infrared band in the spectrum and was applied to the quality evaluation of pc-LEDs in near-infrared applications. The distribution of DR-NIR phosphors as close to the chip as possible can effectively increase the *BRI*_707–830_ by 10.5%.

## Figures and Tables

**Figure 1 micromachines-15-01035-f001:**
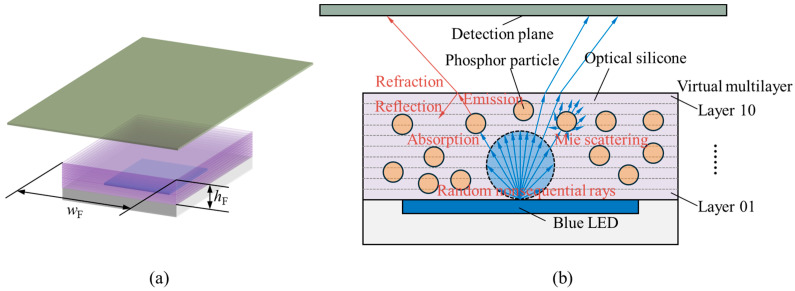
(**a**) 3D schematic diagram of the simulation model; (**b**) 2D schematic diagram of the pc-LED optical model with a 10-layer virtual fluorescent film established by the Monte Carlo ray-tracing method.

**Figure 2 micromachines-15-01035-f002:**
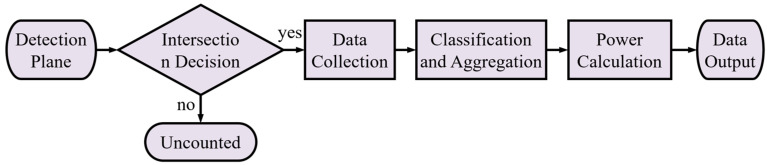
The basic workflow for the detection plane from receiving rays to data output.

**Figure 3 micromachines-15-01035-f003:**
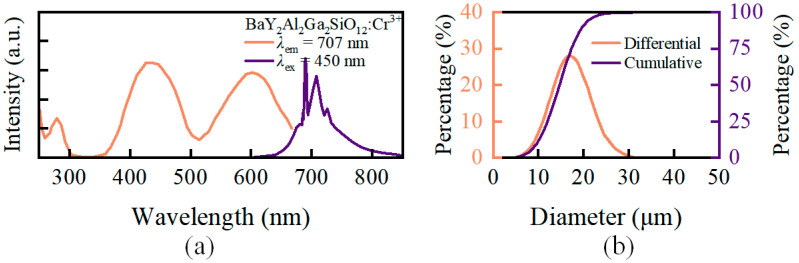
(**a**) PLE and PL spectra of BaY_2_Al_2_Ga_2_SiO_12_:Cr^3+^ phosphor particle [37]; (**b**) Size distribution curves of BaY_2_Al_2_Ga_2_SiO_12_:Cr^3+^.

**Figure 4 micromachines-15-01035-f004:**
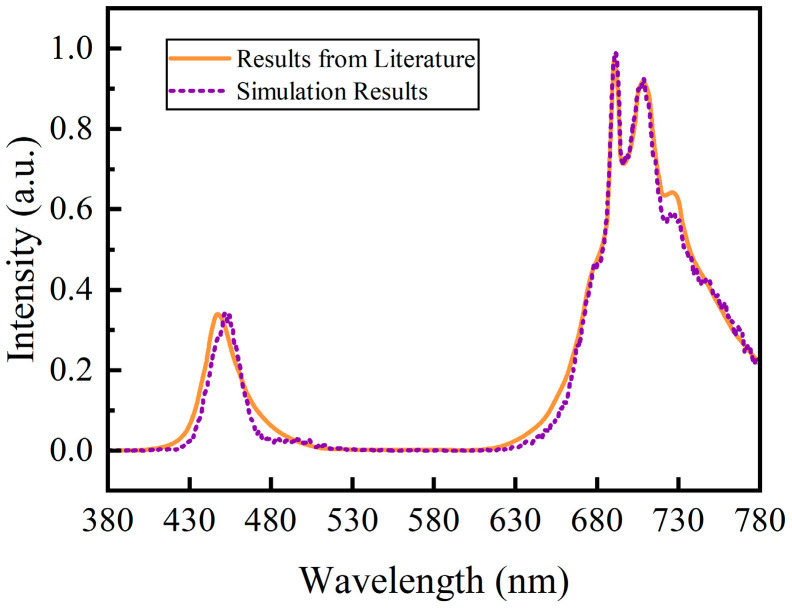
Comparison of validation simulation results of pc-LEDs with a single type of phosphor particle (BaY_2_Al_2_Ga_2_SiO_12_:Cr^3+^) to literature results [37].

**Figure 5 micromachines-15-01035-f005:**
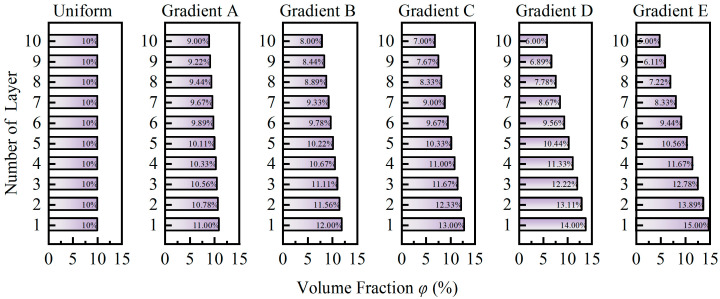
Volume fraction distribution of phosphor particles across the 10 virtual layers. The layer numbers on the y-axis start from the position closest to the blue LED chip, consistent with the schematic in Figure 1.

**Figure 6 micromachines-15-01035-f006:**
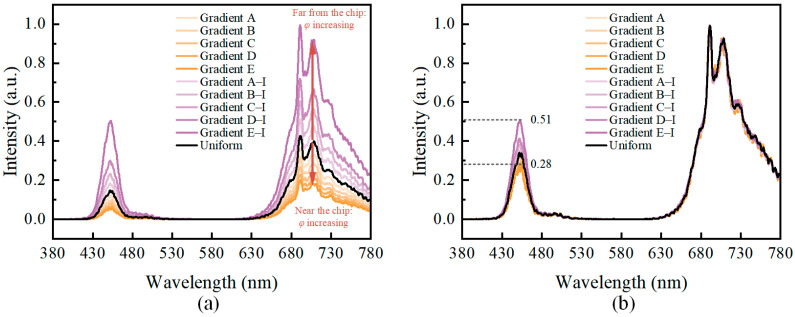
Simulation results with a single type particle. (**a**) Normalized by the overall minimum–maximum values across all groups; (**b**) normalized by the minimum–maximum values for each group individually. The orange curves represent phosphor particles accumulating towards the blue LED chip, while the purple curves represent phosphor particles accumulating far from the chip. The color intensity indicates the relative concentration.

**Figure 7 micromachines-15-01035-f007:**
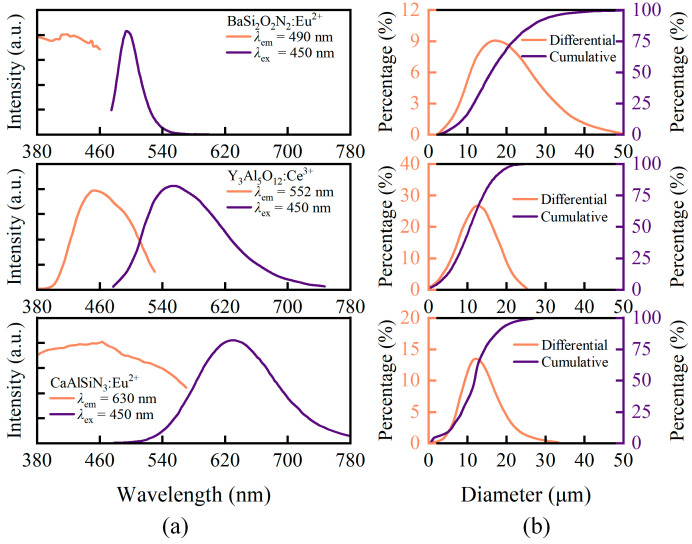
(**a**) PLE and PL spectra of BaSi_2_O_2_N_2_:Eu^2+^, Y_3_Al_5_O_12_:Ce^3+^, and CaAlSiN_3_:Eu^2+^; (**b**) size distribution curves of BaSi_2_O_2_N_2_:Eu^2+^, Y_3_Al_5_O_12_:Ce^3+^, and CaAlSiN_3_:Eu^2+^.

**Figure 8 micromachines-15-01035-f008:**
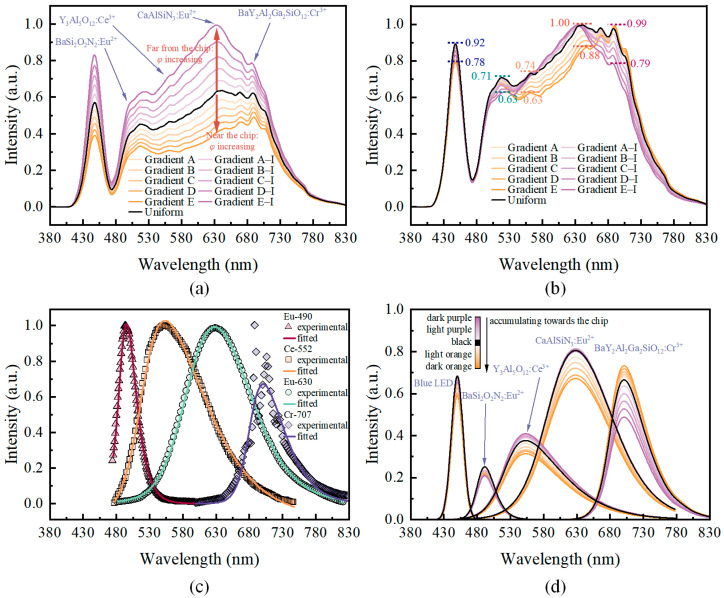
Simulation results with four types of particles. (**a**) Normalized by the overall minimum–maximum values across all groups; (**b**) normalized by the minimum–maximum values for each group individually; (**c**) fitting curves for the four types of phosphors; (**d**) spectral deconvolution peak analysis diagram. The orange curves represent phosphor particles accumulating towards the blue LED chip, while the purple curves represent phosphor particles accumulating far from the chip. The color intensity indicates the relative concentration.

**Figure 9 micromachines-15-01035-f009:**
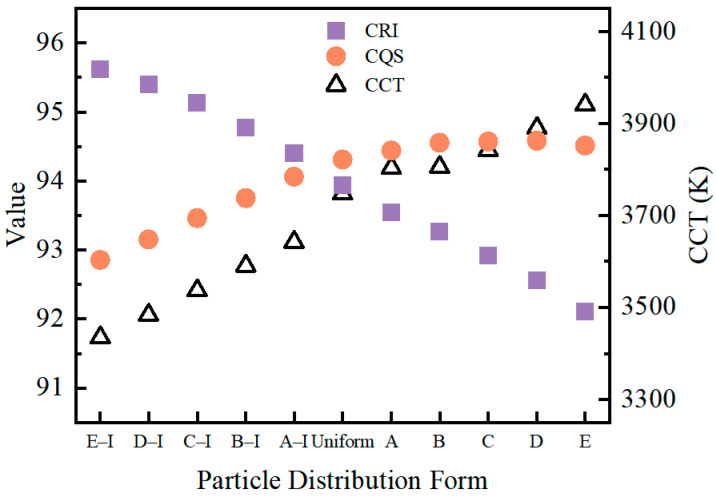
Calculation results for CCT, CRI, and CQS of DR-NIR pc-wLEDs with different particle distributions.

**Figure 10 micromachines-15-01035-f010:**
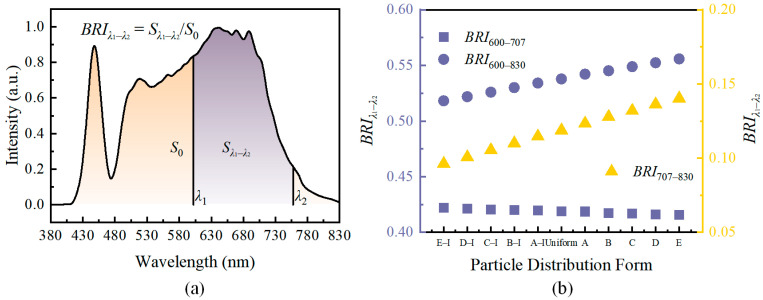
(**a**) Schematic diagram of the principle for defining the Band Ratio Index (*BRI_λ_*_1–_*_λ_*_2_); (**b**) calculation results for *BRI_λ_*_1–_*_λ_*_2_.

**Table 1 micromachines-15-01035-t001:** Parameters and results for fitting peaks of the emission spectra of the four types of phosphor particles.

Phosphors	Parameters					Results	
	*A*	*x* _c_	*w* _1_	*w* _2_	*w* _3_	*RSS*	*R* ^2^
BaSi_2_O_2_N_2_:Eu^2+^	3.7	490	1.0	6.5	12.9	7.7 × 10^−3^	99.9 × 10^−2^
Y_3_Al_5_O_12_:Ce^3+^	1.9	552	61.3	15.3	48.6	2.8 × 10^−2^	99.8 × 10^−2^
CaAlSiN_3_:Eu^2+^	1.5	630	81.4	20.7	34.4	1.0 × 10^−2^	99.9 × 10^−2^
BaY_2_Al_2_Ga_2_SiO_12_:Cr^3+^	1.2	707	32.0	9.8	26.0	5.1 × 10^−1^	93.8 × 10^−2^

## Data Availability

The data presented in this study are available upon request from the corresponding author. The data are not publicly available due to the confidentiality of the project.

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
