# Peer review of "Study on the Performance of Deep Red to Near-Infrared pc-LEDs by the Simulation Method Considering the Distribution of Phosphor Particles"

_micromachines, 2024, doi:10.3390/mi15081035_

Round 1

Reviewer 1 Report

Comments and Suggestions for Authors

In this article, an optical model of NIR pc-LEDs with virtual multilayer fluorescent films was established based on the Monte Carlo ray tracing method. The effects on the spectral power distribution and color quality of NIR pc-LEDs were evaluated by controlling the distribution of phosphor particles in the multilayer films. The work is quite interesting, all the related methods are mentioned and meet the criteria of the journal. I suggest the manuscript be accepted after minor revision.

1. Differental in Figure 3b and 7b should be differential?

2. The author studied the fluorescence intensity, CRI and CCT of the model based on single- and multi-types of fluorescent particles. They explored the model with 10 fluorescent film layers with different particle volume fractions. What is the influence of particle concentration on the fluorescent performance? It would be better if the authors investigated the fluorescent performance with different particle concentrations condition.

3. What about the fluorescence results if all the fluorescent particles distributed in the edge of on the surface of LED packaging?

Reviewer 2 Report

Comments and Suggestions for Authors

The paper is interesting and could have some scientific impact if improved along the lines mentioned below.  Overall, this topic has been well explored in the literature (e.g., see references 20-28) but the contribution of this work is the addition of a new phosphor for pcLEDs.  However, there are several significant errors with this manuscript that must be corrected before publication in any journal.

First, and perhaps most egregious, is the calculation of CRI and CQS values for a device called a NIR pcLED.  CRI and CQS have no meaning for a NIR LED, only for a white LED.  The device described in lines 252 - 354 can be described as a white LED with a deep red phosphor, but it is definitely not a NIR LED.

Second, judging by the emission spectra for the phosphor shown in Figure 3, the phosphor has very little true NIR emissions, but instead could be classified as a deep red emitter.  In general, NIR emitters are of minimal interest in white LEDs because they emitted out the wavelength range of the human eye, and this emission significantly lowers their efficiency.  This is why the incandescent light bulb, which has copious amounts of NIR radiation is considered to be inefficient.

Third, the manuscript conflates results taken from absolute measurements (e.g., Fig. 6a and Fig. 8a) with measurements taken with the local spectral maxima (e.g., Fig. 6b and Fig. 8b).  For the latter, it is ok to say that the peaks within a given spectra are changing relative to each other (i.e., a relative or proportional change) but it should not be implied that there is a change in absolute magnitude.  For example, in lines 234-236, the manuscript points out a relative increase of 82.1% in the 450 nm peak intensity.  But this comparison of only valid for Fig. 6b.  In actuality, this difference is roughly 5X (i.e., 500%) when view from the absolute spectral results in Fig. 6a.  A similar error occurs when discussing Fig. 8.

The manuscript never discusses when the summation limits for the numerator in Eq. 7 only cover the spectral range from 680 nm to 720 nm.  Given the emission profile of the phosphor, this range should be set to a higher ending value.

The manuscript does not disclose the quantum efficiency of the deep red/NIR phosphor that is the subject of this work.  That information must be included in the manuscript since it allows the user to gauge the efficiency of the phosphor and the gauge the absorption from blue, green, and red emitters in Section 3.2.

The discussion of Section 3.2 seemed to only emphasize absorption of the red phosphor by the deep red/NIR phosphor.  According to Fig. 3, the YAG phosphor would also be absorbed to a significant extent (especially the longer wavelength emissions above 530 nm) and there could be some absorption of the blue phosphor.  The only way to properly discuss the spectral changes in Fig. 8b is to deconvolute the spectra according to each phosphor emissions and look at the relative changes between the spectra.

The use of the term "luminous efficiency" (line 40) is invalid for a NIR LED.  Only visible light can have luminous efficiency.

The manuscript has no information on the equipment and procedures used to acquire the data in Fig. 3.  It is essential to the paper to know whether this is an experimental result or a predicted one.

Comments on the Quality of English Language

No issues with the quality of English in this paper.

Reviewer 3 Report

Comments and Suggestions for Authors

Comments on manuscript micromachines-3120255 “Study on the Performance of Near-Infrared pc-LEDs by the Simulation Method Considering the Distribution of Phosphor Particles” by C. Li et al.

Authors of the manuscript report some very interesting results on simulation of pc-LED emission spectra and a dependence of these spectra on particle distribution within converting layer(-s). The results of the study indicate a possibility to predict the emission spectra with high accuracy without extensive experimental studies, those often accompanied by using of expensive chemicals and complex synthesis techniques for phosphor production. However, there are some issues those should be clarified before acceptance to publication of this manuscript. So, I recommend minor revision. Comments and suggestions for improving can be found below.

 1) Authors made an accent on NIR pc-LEDs, but is here any limit to apply the proposed simulation method for pc-LEDs operating in visible region? Moreover, it is shown that the method can be applied to phosphors those emit light in visible region. I suggest removing “Near infrared” from the title of the manuscript to make it more attractive for a wider audience.

2) Authors should describe in more detailed way what fluorescence parameters were used for simulation of emission spectra. The phrase “The phosphor particle parameters used in this part of the simulation experiment were provided by the Changchun Institute of Applied Chemistry.” (lines 261-263) is too general. Does the procedure applied is the same as reported by Tanaka and Puschel (Optics Express, Vol. 32, pp. 2306-2320, 2024)?

3) If emission and excitation spectra in Fig. 3 and Fig. 7 were taken from literature, please indicate corresponding references in the figure captions.  

4) What a reason in Figure 10, Formula 7 and their discussion? Authors discussed only an effect of Cr-707 particles accumulation near LED surface but how about other component of fluorescent film (particles of green, yellow and red phosphors) are their distributions uniform in this case? Please clarify. Maybe it is better to make such discussion for case of single type of phosphor?

5) lines 121-122: it is said “…a central wavelength of 450 nm (μ = 450 nm) and full width at half maximum 121 of 18 nm (σ = 7.64 nm).” Does FWHM in 18 nm correspond to FWHM of blue emission band of LED chip used in experiment?

6) line 160: it is said “n is the density of the particle number”. Does it mean n is particle concentration? Please clarify.

7) line 178: it is said “The reflecting cup structure was not defined, and this consistent setting across all simulation groups did not compromise the significance of horizontal comparisons”. What authors tried to state in this sentence? The last part is very hard to understand. Please clarify.

8) check and correct a journal name in ref 37

Comments on the Quality of English Language

Some typos and wordings should be corrected, e.g. "spectras" at caption of Fig. 7 and "size distribution curves" instead of "size distribution maps"

Round 2

Reviewer 2 Report

Comments and Suggestions for Authors

I appreciate the authors of this manuscript responding to my previous critique, and I think the manuscript has been greatly improved both from a technical content and a scientific soundness perspective.  There are still several issues that have not been adequately addressed in the response and should be corrected before the paper is published.

First, the explanation given for the numerator summation limits in Eq. 8 is inadequate, and the summation range appears to be chosen at random by the authors.  As shown in Fig. 4, the emission range of the Cr3+ (i.e., deep red) phosphor is 630 to > 780 nm.  I would expect the summation limits of the numerator of the SDR-NIR factor to be something similar to this and not just arbitrarily truncated at 680 and 720 nm.  Granted, consideration has to be given for the red-orange phosphor (CaAlSiN3:Eu2+) which partially overlaps with the deep red phosphor especially at lower wavelengths (630 nm to ~ 700 nm).  I am especially surprised that a factor that is now called SDR-NIR does not have the summation range extend into the NIR where there would be minimal overlap with the red-orange phosphor.  This would give a more accurate representation of deep red and NIR emissions.

Second, the spectral deconvolution is Fig. 8c is a nice addition to the manuscript and strengthens the work.  However, this deconvolution was built using assumed symmetric Gaussian fits, and there is an error introduced with a symmetrical Gaussian fit because the phosphor emissions are not always symmetrical.  For example, the YAG emission spectra in Fig. 7 is clearly asymmetric.  Perhaps other fits such as the logistic function or asymmetric Gaussian would provide greater accuracy.  See the following references: 

1.       Fraser, R.D.B., Suzuki, E. (1969). Resolution of overlapping bands. Functions for simulating band shapes. Analytical Chemistry, 41, 37–39.

2.       Reifegerste, F., Lienig, J. (2008). Modelling of the temperature and current dependence of LED spectra. Journal of Light & Visual Environments, 32, 288–294.

3.       Jeong, S.S., Ko, J.-H. (2012). Analysis of the spectral characteristics of white light-emitting diodes under various thermal environments. Journal of Information Display, 13(1), 37-42.  For this type of curve fit, you should also include a measure of the accuracy of the fit (such as mean square error) and provide a measure of how each Gaussian fit matches up with the actual emission spectral.

As a minor point, the findings in the new Fig. 8c show a clear dependence on Cr3+ emission depending upon the location within the phosphor structure.  There are also variations in the blue (BaSi2O2N2:Eu2+), green (i.e., YAG), and red-orange phosphors emissions depending upon the location of the CR3+ phosphor due to reabsorption.  This finding is potentially interesting and should be discussed further in the manuscript.  

Round 3

Reviewer 2 Report

Comments and Suggestions for Authors

The latest additions to the manuscript have greatly improved the content.  I have three small suggestions.  First, in Line 354, the manuscript references Figure 7(b) when I think you mean Figure 8(d).  Be sure to double check the figure references in the final copy since some the of figures have changed.  Second, in line 357 you reference that the Mie geometric scattering coefficient of the particles playing a role.  While this may be true, I think the differences are probably small.  In order to make the point that Mie geometric factors are important, you would have to list the Mie backscattering coefficients for each particle to show there is a significant difference.  I don't think the paper needs that.  I think the trends you are observing are actually due to photon reabsorption as you correctly identified in the new text.  Finally, in line 363, the manuscript states that the amplitude of variation of the Ce-552 phosphor (i.e., green) is "not very significant."  I think this is a debatable, but it is probably sufficient to say that it is "not as significant" as the Eu-630 (i.e., red) phosphor.
